# Comment on "A comparison of catchment travel times and storage deduced from deuterium and tritium tracers using StorAge Selection functions" by Rodriguez et al. (2021)

Michael Kilgour Stewart[1], Uwe Morgenstern[2], Ian Cartwright[3]

[1]Aquifer Dynamics and GNS Science, PO Box 30 368, Lower Hutt 5040, New Zealand
[2]GNS Science, PO Box 30 368, Lower Hutt 5040, New Zealand
[3]School of Earth, Atmosphere and Environment, Monash University, Clayton, Vic 3800, Australia

*Correspondence to*: Michael Stewart (m.stewart@gns.cri.nz)

**Abstract.** The combined use of deuterium and tritium to determine travel time distributions (TTDs) in streams is an important development in catchment hydrology (Rodriguez et al., 2021). This comment takes issue with Rodriguez et al.'s assertion that the truncation hypothesis may not hold for catchments in general, i.e. that the use of stable isotopes alone may not lead to underestimation of travel times or storage compared to tritium. We discuss reasons why the truncation hypothesis may not appear to hold for the catchment studied by Rodriguez et al. (2021), but could still apply to the majority of catchments. We also discuss more generally future applications of tritium in northern and southern hemisphere catchments.

## 1 Introduction

Rodriguez et al. (2021) applied deuterium ($^2$H) and tritium ($^3$H) measurements to determine transit time distributions (TTDs) in a forested headwater catchment (the Weierbach Catchment in Luxembourg). They used the method of StorAge Selection (SAS) functions (Botter et al., 2011; Van Der Velde et al., 2012; Benettin et al., 2017) to transform the input (rainfall) values of the two tracers to match the concentrations of the tracers in the stream water draining the catchment also taking account of output via evapotranspiration. Then they tested the (truncation) hypothesis that the tritium TTD would extend to longer transit times than the deuterium TTD. They found that the TTDs were not different within error and concluded that "the stable isotopes do not seem to systematically underestimate travel times or storage compared to tritium".

The truncation hypothesis put forward by Stewart et al. (2010) states: "The use of stable isotope tracers ($^2$H and $^{18}$O) and chloride (Cl) more than any other tool has influenced the development of the field since their first use in the 1970s (Dinçer et al., 1970). […] But what if the information gleaned from stable isotopes actually biased our understanding of how catchments store and transmit water? What if our now, almost exclusive use of stable isotopes has led us down a pathway that has skewed our view of streamwater residence time? Here we show [using tritium ($^3$H)] that deeper groundwater contributes more to streamflow than we are able to ascertain using conventional stable isotope–based hydrograph separation and streamflow residence time approaches."

In this comment we argue that the truncation hypothesis is not generally invalidated by the Rodriguez et al. (2021) study.

**2 The Weierbach Catchment study**

The conclusion that no significant old water (beyond the range that can be resolved by stable isotopes) was identified by $^3$H in the Weierbach Catchment stream does not mean that such old water does not exist in other catchments and therefore that the truncation hypothesis should be rejected for all catchments. In fact, there is evidence that such old water is present in many other catchments (e.g. Table 1). There are at least two possible reasons for the results from the Weierbach Catchment:

1. The large majority of water may actually be within the TT range that can be determined by stable isotopes. In that case, analysis of the $^2$H and $^3$H isotope data would be expected to yield similar TTDs. Evidence that this may be the case is given by:

    a. The physical characteristics of the catchment. The Weierbach Catchment is small (42 ha) and has a thin layer of <1.5 m of porous gravels overlying variably-weathered bedrock extending to about 5 m depth (Pfister et al., 2017). The catchment area of $4.2 \times 10^5$ m$^2$ and annual rainfall of about 0.95 m imply water input of $4 \times 10^5$ m$^3$ of which about 50% or $2 \times 10^5$ m$^3$/year runs off. If the porosity of the weathered bedrock is 0.1 to 0.2, the water in storage will be $2.1 - 4.2 \times 10^5$ m$^3$. This gives an expected MTT (calculated from storage / runoff) of 1-2 years. If most of the water comes from the thin gravels, the MTT will be shorter. Either way, it seems unlikely that there is much water with long TTs in this catchment, although this does not, of course, rule it out.

    b. Both $^2$H and $^3$H yield a mean transit time (MTT) of about 3 years and a 90$^{th}$ TT percentile of about 5 years at Weierbach Catchment (Rodriguez et al., 2021). Neither isotope gives any indication of a substantial presence of older water.

2. The use of tritium for determining long TTs (or long tails) in parts of the Northern Hemisphere is still compromised by the presence of bomb tritium which partially masks the effect of radioactive decay (see below). This means that even if there was older water present in the Weierbach Catchment, tritium from a short period of sampling would not be very effective for identifying it (as shown below).

**3 The current situation of tritium in precipitation**

Northern Hemisphere (NH) precipitation was much more strongly affected by bomb tritium due to nuclear weapons testing in the 1950s than Southern Hemisphere (SH) precipitation (as shown by data in the WISER database of the International Atomic Energy Agency, IAEA and WMO, 2020). Fig. 1 shows the tritium records in NH precipitation from Trier (Germany) and Oregon (USA), and SH precipitation from Kaitoke (New Zealand). It can be seen that the NH bomb peak was about 100 times bigger than the SH bomb peak and was about two years earlier. The Trier record (Fig. 1a) is from Schmidt et al. (2020) with earlier records from Vienna, the Oregon record (Fig. 1b) is from Michel (2006) with scaled earlier and later records (Michel, 2006, Stewart et al., 2012), and the Kaitoke record (Fig. 1c) is from Morgenstern and Taylor (2009), and later data). The Trier record is the nearest to the Weierbach Catchment, and the Oregon record is similar to that at Southern Sierra Nevada on the west coast of the United States (Visser et al., 2019).

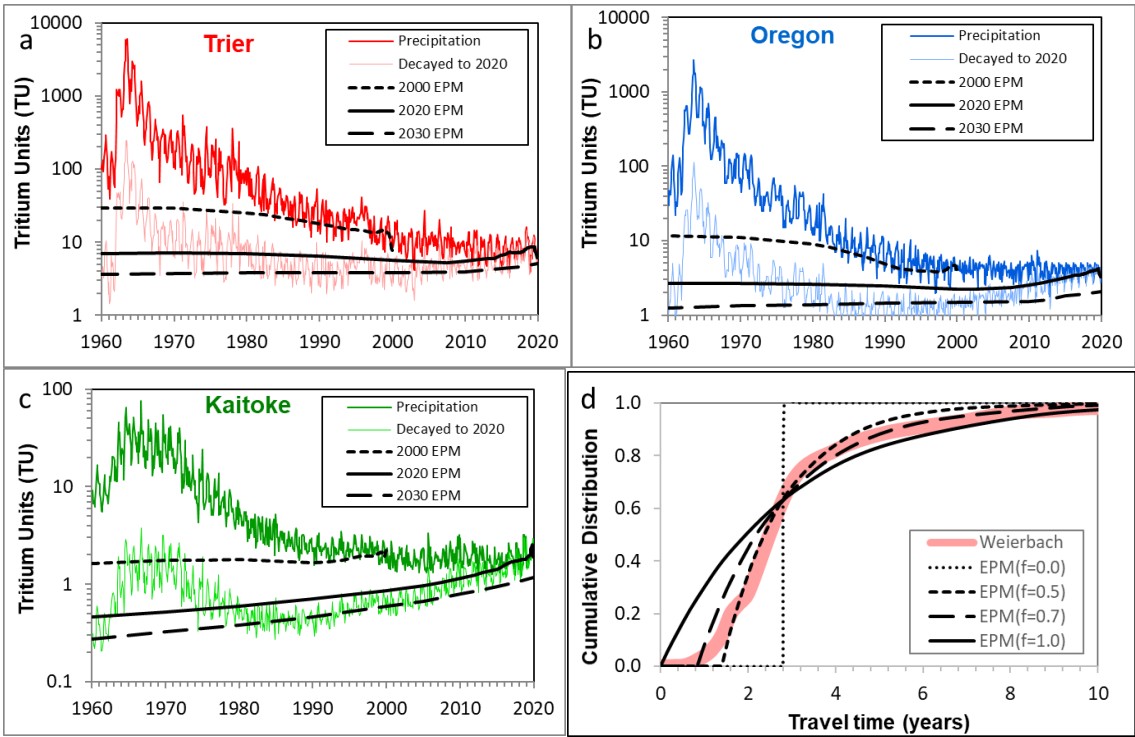

Figure 1. Tritium concentration in precipitation at (a) Trier, Germany, (b) Oregon, USA, and (c) Kaitoke, NZ, after accounting for radioactive decay up to 2020, and after mixing of the decayed concentrations using an EPM(f = 0.7) model (black curves). (d) Comparison of flow-weighted cumulative stream distribution from Weierbach Catchment (Fig. 7c, Rodriguez et al., 2021) with cumulative EPM curves.

The figures also show the reductions in $^3$H activities of the rainfall due to radioactive decay from the rain date to 2020 (curves marked "decayed to 2020"). Due to radioactive decay, the present-day $^3$H activities of older water are lower than current rainfall except when they are affected by bomb tritium which peaked in the 1960s.

However, water present in a stream or spring will have a wide spread of ages due to mixing during flow in the soil and country rock. This mixing is simulated here by using the exponential piston flow lumped parameter model (EPM) with an exponential fraction f = 0.7 (Maloszewski and Zuber, 1982). The black mixing curves in Figs. 1a-c show the variations in $^3$H expected for samples collected in 2000, 2020 and 2030, with the MTT represented by the year (e.g. a 2020 sample $^3$H result yielding MTT=10 years is plotted at 2010). The EPM(f = 0.7) model is chosen here as a realistic mid-range estimate between the model with no mixing (EPM(f=0)), also called the piston flow model that would produce the radioactively decayed curves in Fig. 1) and the fully mixed model (EPM(f=1)), also called the exponential model). Fig. 1d compares the flow-weighted cumulative stream distribution from Fig. 7c of Rodriguez et al. (Weierbach) with cumulative EPM curves, the EPM(f=0.7) curve is quite similar to the Weierbach curve. Other lumped parameter models such as the gamma (GM) or dispersion (DM) models could also have been used; the GM(α=3) and DM(D$_P$=0.22) models produce distributions that are closely equivalent to the EPM(f=0.7) curve (Stewart et al., 2017). Cartwright et al. (2018) also showed that numerical models that include dispersion yield similar curves; such models are independent of the lumped parameter model approach.

The EPM model curves for Trier (Fig. 1a) are relatively flat because of the combined effects of radioactive decay and mixing, but the rising trend with MTT in the 2000 curve is gradually changing to a falling trend with MTT in the 2030 curve. At present (2020), water with an MTT of 0 years has a mean tritium activity of about

8.2 TU, and this falls gradually to 5.5 TU for water with an MTT of 10 years. Water with greater MTTs then has slightly increased tritium activities with a maximum of 7.1 TU for water with MTTs of 50-60 years, due to the presence of remnant bomb water with higher tritium activities. This means that there is a capacity for detecting water with MTTs of 0–10 years because of the difference in $^3$H activities of up to 2.7 TU, but water with longer MTTs will not be distinguishable from younger water with tritium at present. The precipitation record also shows a significant seasonal variation (approximately 8 TU peak-to-peak) due to leak of tritium from the stratosphere each spring. This has not previously been used for dating until Visser et al. (2019) and Rodriguez et al. (2021), however, the fact that this variation is seasonal and considerably larger than the long-term variation expected from the mixing model suggests that tritium will give results similar to those from the stable isotope seasonal variations, but with a small bias towards longer MTTs because of the small decrease of tritium activity between waters with MTTs of 0 and 10 years (in 2020) due to radioactive decay.

Another study that determined the TTD using tritium measurements via the method of StorAge Selection functions was that of Visser et al. (2019) in the Southern Sierra Nevada on the west coast of the United States. They also applied the method to $^{18}$O, $^{35}$S, and $^{22}$Na. Their reconstructed tritium input (their Fig. S4.1, represented here by the Oregon record (Fig. 1b)) shows the high NH bomb peak similar to the Trier input in Fig. 1a, but tritium concentrations in rainfall fell more rapidly after the tritium bomb pulse because of the influence of moisture from the Pacific Ocean (as shown by Michel (2006) and Stewart et al. (2012)). Visser et al. found that tritium in precipitation was flat from 2000 to 2020, and therefore that radioactively decayed tritium reached its lowest level in 2000 before rising because of the high NH bomb pulse at greater MTTs. This means that tritium can be used more effectively at present for identifying older water (with MTTs up to 20 years) in the southern Sierra Nevada and west coast of the United States than in western Europe, but the 1960s bomb pulse is a problem for both. However, perhaps 86% of catchments contain water with MTTs less than 20 years (see estimate below from Stewart et al., 2010). The reconstructed southern Sierra Nevada record also has a substantial seasonal variation of 9.5 TU peak-to-peak.

In summary, tritium concentrations currently have limited capacity to reveal old waters in NH catchments, compared to the SH, because of the presence of the remnant bomb tritium pulse from the 1960's. In 2000, tritium was even less conclusive as an age tracer in streams (Fig. 1). This is about when high-frequency stable isotope studies became accessible, and were applied in NH high-precipitation, low-ET catchments. The collective understanding of watershed response times may have been influenced by the availability of such data during these decades. It is still difficult to see older water using $^3$H now (especially when only short periods of measurements are available), however, post-bomb pulse conditions are considerably different in different parts of the NH (as shown by the different shapes of the tritium input functions for Trier and Oregon). The situation is gradually improving because bomb tritium is decaying and becoming further away in time. Samples collected now will be valuable in combination with samples collected in the future but relatively long series (up to a decade) may be necessary. The 2030 EPM curve for Trier (Fig. 1a) shows the behaviour that would be expected for samples collected in 2030 for central Western Europe. $^3$H activities would decrease from 8.2 to 3.8 TU for water with MTTs up to 20 years, but then the curve would be flat at 3.8 TU for longer MTTs. This will give better

potential for identifying water with MTTs up to 20 years however. For Oregon, the 2030 curve will be falling monotonically with MTT which will give good potential for dating.

In contrast, for waters collected in 2020 in Kaitoke in the SH (Fig. 1c) there is a simple decline in $^3$H activities with MTT (from 1.9 TU at 0 years to 0.46 TU at 60 years) and the decrease continues to higher MTTs (not shown). This means that the determination of water TTs is only limited by analytical techniques (and is conservatively 100 years). The precipitation record also shows a seasonal variation of 1 TU peak-to-peak which affects the EPM model curve at young MTTs and can in principle be used to determine young MTTs by the smoothing effect as

with stable isotopes. Fig. 1c also shows the 2000 EPM model curve, which was almost flat after an initial drop, and the 2030 EPM model curve, which will be a steeper version of the 2020 curve.

## 4 Evidence that $^3$H TTDs can be different from $^2$H TTDs

After the bomb tritium pulse in the 1960s, NH precipitation became enriched with tritium which could be in principle, and in fact was, effectively traced through hydrological systems (Fig. 1a). Tritium studies from this

period (1960–1990) showed that the TTDs of catchments covered a wide range of values (cited in Stewart et al., 2010). Estimated MTTs ranged from less than one year to decades (e.g. of the catchments listed in Stewart et al. (2010), 45% had tritium MTTs of 0-5 years, 41% had MTTs of 5-20 years and 14% had MTTs greater than 20 years). Because MTTs deduced from stable isotope/chloride measurements are based on short-term (seasonal) variations which are partially or fully attenuated after about five years depending on the mixing model applied,

MTTs greater than about five years are more difficult to substantiate (Stewart et al., 2010). If the MTTs were in fact longer than five years, they would nevertheless tend to look like about five years with stable isotopes depending on the model applied. So a large fraction of the catchments (up to 55%) would be expected to have had different MTTs when evaluated with tritium or with stable isotopes. This was supported by the study of Seeger and Weiler (2014) who analysed stable isotope data on 24 mesoscale Swiss catchments. They stated, "Given a

sufficiently high measurement frequency, stable water isotope data should be suited to characterise the short term and intermediate part of a catchment's TTD, but it certainly does not contain enough information to determine complete TTDs or actual MTTs of a catchment."

The SH reached the stage of having problems with interpretation of tritium synchronously with the NH, but the

problem began to decrease much sooner in the SH (about 1990, see Fig. 1c) because of the c100 times smaller bomb pulse in the SH. The current situation allows for the effective use of tritium for estimating TTDs in SH catchments and the environment continues to become more favourable for the use of tritium.

A recent summary of tritium results from Australian catchments illustrates the capacity of tritium to identify old

water in SH catchments (Table 1, Cartwright et al., 2020; Duvert et al., 2016). These MTTs are much longer than could be estimated by stable isotopes, so they show that truncation is very much an issue. The measurements cover both baseflow and high flow conditions (e.g. the Ovens Catchment measurements spanned Q8 to Q85 flows, and the LaTrobe and Gellibrand measurements Q10 to Q95 flows), implying that the truncation issue applies to both baseflow and high flows.

Table 1: MTTs for Australian catchments based on tritium

| Catchment | Baseflow (years) | High flow (years) |
|---|---|---|
| Ovens[1] | 9 - 30 | 4 - 10 |
| Yarra[1] | 20 - 50 | 13 - 37 |
| LaTrobe[1] | 18 - 41 | 7 – 19 |
| Gellibrand[1] | 14 - >100 | 7 – 20 |
| Deep Creek[1] | 2 - 6 | <1 – 38 |
| Lyrebird[1] | 45 - 50 | 9 - 10 |
| Teviot Brook[2] | 17±6 | 38±15 |

[1] Cartwright et al. (2020), [2] Duvert et al. (2016)

Rodriguez et al. (2021) have suggested that the differences between tritium and stable isotope MTTs in the literature could be attributed to methodological problems with the use and interpretation of tritium. Methodological problems they identified included:

1. Sampling differences in streams - tritium concentrations have typically been determined at far less frequency and flow diversity (often only at baseflow) than stable isotopes, whereas stable isotope studies are becoming increasingly intensive and average over the full range of streamflows. However, that is usually not the case for New Zealand tritium studies (e.g. Morgenstern et al., 2010, Morgenstern et al., 2015) and the Australian studies cited above that have estimated MTTs at a range of streamflows.

2. TTD assumptions - tritium studies have often assumed steady-state TTDs based on analytical expressions, whereas more recent stable isotope studies have used time-variable TTDs which cannot be expressed analytically. But a steady-state TTD applied separately to individual tritium samples in a time sequence can allow the TTD parameters to vary from sample to sample and therefore can yield a time-variable TTD as demonstrated by Morgenstern et al. (2010).

3. Methodology - tritium studies have used simpler mathematical formulations not involving antecedent rainfall and evapotranspiration, whereas stable isotope studies are now more commonly incorporating these elements via StorAge Selection function methods. However, recharge models have been applied to tritium studies to account for antecedent rainfall and evapotranspiration (e.g. Stewart et al., 2007; Morgenstern et al., 2010).

While these observations are valid for some tritium studies, they do not alter the fundamental point that tritium in the right conditions can identify the presence of older water whereas stable isotopes cannot. Because tritium decays, the MTT can be indicated by reduction in tritium activity over time and not just by attenuation of seasonal tracer variability in the input signal.

Both stable isotopes and tritium can be affected by aggregation effects causing them to underestimate the MTT in heterogeneous catchments (Kirchner, 2016; Stewart et al., 2017). The effects will be similar for both if MTT estimation is based on smoothing of seasonal variations, but different if the stable isotope MTT estimation is based on smoothing of a seasonal variation and the tritium MTT estimation based on radioactive decay. The latter causes underestimation on a much longer time scale.

**5 Discussion and Conclusions**

The situation in both hemispheres has changed in time as bomb pulse tritium has worked its way through the systems. In the NH, elevated tritium values were useful for determining TTDs for a few decades past the bomb peak (1960s to 1980s). Then as the bomb tritium pulse decayed, single tritium measurements gave unclear and ambiguous MTT results, so that series of measurements were needed, and still are. But the NH is slowly emerging from this period. In the SH, this ambiguous period was shorter and tritium is now effective for determining long TTs in catchments. Because estimates of MTTs can be made from single tritium measurements when bomb tritium is at a low level (provided the form of the TTD can be reasonably determined), the issue of time-variability is different for $^3$H and $^2$H. Series of measurements are always needed for $^2$H.

In addition to the long-term variations of tritium concentrations due to the bomb tritium pulse and radioactive decay, tritium shows medium-term, short-term (seasonal) and probably even shorter-term (e.g., between different rainfall events) variations. These are largely unexplored (as pointed out by Rodriguez et al. (2021)). The seasonal variations are increasingly prominent because tritium in precipitation is at background levels (except where tritium from local nuclear industry is present). This seasonal and event-scale tritium variation contains the same information as the variation of stable isotopes. Indeed Rodriguez et al. (2021) have shown that tritium is as effective as stable isotopes when used to determine TTDs from seasonal fluctuations. However, we feel that the real strength of tritium in comparison with stable isotopes is for determining longer TTs in catchments; at present and for some time into the future in the NH, this will require widely spaced measurements in time. For this reason, it seems unreasonable to us to consider only very detailed sampling of tritium on the scale of stable isotopes over short periods. Instead our recommendation is that more widely spaced measurements in time are included, as well as covering the full range of flow conditions so that results are not biased to low flows.

The difference between the tritium input functions in the NH and SH with regard to remnant bomb tritium in hydrological systems suggests that it would be valuable to carry out a study like that of Rodriguez et al. (2021) (or Visser et al., 2019) in the SH to test the comparative merits of tritium and stable isotopes for identifying the full range of travel times in catchments. Would stable isotopes show the full extent of the long tail in comparison with tritium as asserted by Rodriguez et al. (2021) or would tritium show a 'heavier' long tail as asserted here?

*Data availability*. No new datasets were developed or used in this comment.

*Author contributions*. MKS prepared the first draft after discussion with UM. Substantive additions and edits were made by IC.

*Competing interests*. The authors declare that they have no conflict of interest.

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
