# Peer review of "Comment on "A comparison of catchment travel times and storage deduced from deuterium and tritium tracers using StorAge Selection functions" by Rodriguez et al. (2021)"

_Hydrology and Earth System Sciences, 2021_

## Author Response (AR1)

Dear Dr Mariano Moreno De Las Heras,

Thank you. The changes made are detailed below. The line numbers given below refer to the marked up version.

Reply to Francesc Gallart (Reviewer #1, R1)

R1: My opinion is that the Stewart et al. "Comment" is opportune as it provides adequate and valuable scientific discussion following the commented article.

Authors: We thank Reviewer #1 for constructive evaluation of our work.

Changes: No change needed.

R1: Nevertheless, I had some questions while reading it, and therefore I suggest some small changes.

- The article by Rodriguez et al., (2021) does not claim a "general rejection of the truncation hypothesis" but it is true that a quick diagonal reading of that paper might give the reader this impression. The "Comment" should be more consistent with the actual content of the commented article.

Authors: We agree that we may have overstated the case here, and will clarify this statement. We were drawing attention to the fact that the Weierbach Catchment may not be the ideal catchment in which to test the truncation hypothesis.

Changes: L10-14 The wording now is "This Comment takes issue with Rodriguez et al.'s assertion that the truncation hypothesis may not hold for catchments in general, i.e. that the use of stable isotopes alone may not lead to underestimation of travel times or storage compared to tritium". Wording has also been changed elsewhere in the Comment for this reason.

R1: The "Comment" makes sometimes a use of the TTD concept that, in my opinion, is confusing and MTT should be used instead in several instances. Strictly, the TTD term refers to the shape of the distribution, which cannot be obtained from a single tracer observation. Traditional use of single tracer dating needs the assumption of a given TTD (usually represented by a mixing model), so that a subsequent MTT can be estimated (Maloszewski and Zuber, 1982).

Authors: Agreed, we will revise our use of TTD and MTT.

Changes: We changed TTD to MTT in L221, 224, 242.

R1: The caption and the corresponding text of Fig. 1 (incorrectly indicated as Fig. 1a) are not sufficiently clear. At least the text "The blue mixing curves show the variations in $^3$H with MTT expected for samples collected in 2020 and 2030." should be substituted by something like "The blue mixing curves show the variations in $^3$H expected for samples collected in 2020 and 2030, with the MTT represented by the year."

Authors: Agreed, we will revise this.

Changes: We changed the caption (L88-92) and text (L99-101). The text is now "The black mixing curves in Figs. 1a-c show the variations in $^3$H expected for samples collected in 2000, 2020 and 2030, with the MTT represented by the year (e.g. a 2020 sample $^3$H result yielding MTT=10 years is plotted at 2010).

R1: The fact that the work by Visser et al. (2019) was conducted in the Southern Sierra Nevada should be stated closer to the beginning of its mention, to make easier the understanding of the differences between the tritium inputs there and in Trier.

Authors: Agreed.

Changes: We now state this in L128-129.

R1: By the end of section 4, some short comment on the different roles of aggregation effects on TTDs when stable isotopes and tritium are used, based on Kirchner (2016) and Stewart et al. (2017), would be welcome.

Authors: Agreed, we will do this. Aggregation effects cause MTTs to be underestimated by both stable isotope and tritium measurements if the catchment is heterogeneous. The younger (of two components) has a disproportionate effect.

Changes: We inserted a new paragraph describing the effects of aggregation of spatially heterogeneous catchments on MTTs (L215-219).

R1: References

Maloszewski, P. and Zuber, A.: Determining the turnover time of groundwater systems with the aid of environmental tracers: 1. Models and their applicability, J. Hydrol., 57, 207–231, https://doi.org/10.1016/0022-1694(82)90147-0, 1982.

Stewart, M. K., Morgenstern, U., Gusyev, M. A., and Maloszewski, P.: Aggregation effects on tritium-based mean transit times and young water fractions in spatially heterogeneous catchments and groundwater systems, Hydrol. Earth Syst. Sci., 21, 4615–4627, https://doi.org/10.5194/hess-21-4615-2017, 2017.

Visser, A., Thaw, M., Deinhart, A., Bibby, R., Safeeq, M., Conklin, M., Esser, B., and Van der Velde, Y.: Cosmogenic Isotopes Unravel the Hydrochronology and Water Storage Dynamics of the Southern Sierra Critical Zone, Water Resour. Res., 55, 1429– 1450,  ttps://doi.org/10.1029/2018WR023665, 2019.

Reply to Nicolas Rodriguez (CC1)

CC1: We thank Stewart et al. for engaging in a discussion on our contribution. We also thank Francesc Gallart for contributing to this discussion. Here, we take the opportunity to clarify and re-iterate on several key points.

Authors: We thank Nicolas Rodriguez for constructive evaluation of our work.

Changes: No changes needed.

CC1: While we agree with Stewart et al. that tritium is an extremely useful tracer in catchment hydrology, with a high information content (in the case of Rodriguez et al. 2021, higher than stable isotopes), we believe that the reasoning put forward in the comment is flawed. Indeed, in the comment, catchments (as highly dynamic systems) are treated as systems at steady state. Thus, the finding of different travel times can simply be an artifact of the mathematical approach chosen by Stewart et al. because the latter is nowadays known to be a hindrance to deriving more realistic travel times.

Authors: We agree that the assumption of steady state sometimes used with tritium measurements is not ideal for catchment systems. However, we think that even if the two isotopes were treated in the same way experimentally and mathematically (as done by Rodriguez et al., 2021), we would still expect different travel time distributions if the stable isotope input varied seasonally, and the tritium input had both seasonal and radioactive decay variations provided there was enough old water in the stream to reveal the difference. There is no a priori reason to necessarily assume steady state conditions (at least in the SH) because tritium is radioactive. If there are radioactive variations then tritium measurements at various flows allows identification of non-steady state flow conditions.

Changes: This is a general statement by Rodriguez (CC1), which we argued with above. No specific changes were made to our Comment because of this argument.

CC1: We would like to clarify on the claim "that no significant old water (beyond the range that can be resolved by stable isotopes) was identified by 3H in the Weierbach Catchment stream does not mean that such old water does not exist in other catchments". We agree that old water contributions exist in other catchments, and that those can be older than the ones observed in the Weierbach. However, we disagree with the notion that different travel times are identified with both tracers, as this notion lacks any physical process-based explanation, and it can most likely be explained by limitations in the previously applied age-dating models for stable isotopes and tritium.

Authors: This is a restatement of the argument above. We do feel that there is a physical explanation (namely radioactive decay), although this tends to be masked at present by remnant bomb tritium with the Trier tritium input function (as explained in the Comment).

Changes: This continues the argument above. Another part of the physical explanation is that mixtures of water parcels with different MTTs in stream samples acts to remove the input signals for both $^2H$ and $^3H$. No specific changes were made to our Comment because of this argument.

CC1: With increasing mean/median travel times and contributions of old water in streamflow, we can surely assume that the identifiability of model parameters solely based on stable isotopes becomes increasingly difficult.

Authors: Agreed.

Changes: No change needed.

CC1: However, the travel time of a water parcel physically has only one median age, and the parameters calibrated with stable isotopes can still suggest the presence of old water even if they are not clearly identifiable due to increased uncertainties.

Authors: This argument is not clear. Streamflow measurements involve mixtures of many water parcels and therefore have age distributions. Mixture of waters with different ages leads to attenuation of input patterns which are the basis of the stable isotope age determination method. This attenuation makes old water less visible to the method. The same could be said for tritium if the effects of radioactive decay are counteracted by remnant bomb tritium (in the NH). If not, the decrease with time due to radioactive decay should make old water more visible to tritium.

Changes: No change needed.

CC1: Furthermore, in the framework of time-varying travel times derived through SAS functions, the discussion on the differences between SH and NH tritium concentrations in precipitation is not completely relevant. We did not need the current tritium values in the stream to be very different from the water that was recharged decades ago. As explained in the discussion section of Rodriguez et al. (2021), we did not only rely on tritium radioactive decay to age-date water. The method we used can accommodate any tritium input signal. The tritium input signal will, however, affect how much information on water ages can be extracted with the method (we showed how this information can be quantified). We simply accounted for radioactive decay, and the decay likely allows identifying model parameters more precisely compared to stable isotopes for catchments with longer travel times and larger contributions of old water.

Authors: In fact, as shown in the Comment there is at present little radioactive decay component of the tritium input variation at Weierbach Catchment because it is masked by lingering bomb tritium, there is only the seasonal variation matching the seasonal variation in the stable isotope input record. So the tritium and stable isotopes would be expected to give similar travel times. This issue of lingering bomb tritium makes it difficult at present to test the truncation hypothesis generally with the Trier tritium input record.

Changes: No specific changes were made to our Comment because of this argument.

CC1: That being said, it would be interesting to evaluate whether the information content in the tritium input is higher in the SH compared to the NH. This can be tested in a more robust way than in this comment, and we invite Stewart et al. to apply our framework to test whether their suggestion holds against data.

Authors: We agree that this would be a valuable thing to do. However, this is beyond the scope of this comment and we do not currently have suitable SH datasets.

Changes: We have inserted a paragraph at the end proposing the suggested study. L251-256 "The difference between the tritium input functions in the NH and SH with regard to remnant bomb tritium in hydrological systems suggests that it would be valuable to carry out a study like that of Rodriguez et al. (2021) (or Visser et al., 2019) in the SH to test the comparative merits of tritium and stable isotopes for identifying the full range of travel times in catchments. Would the stable isotopes show the full extent of the long tail in comparison with the tritium as suggested by Rodriguez et al. (2021) or would tritium show a 'heavier' long tail as suggested here?"

CC1: We do not support the recommendation in the comment by Stewart et al. to keep sampling tritium sparsely over longer periods. This will very likely bias the tritium data towards hydrological recessions, which by definition will more likely contain older water. Also, this will most likely entirely miss the short-lasting events associated with younger water. We want to re-iterate here that findings from early work with tritium showed the potential of tritium for revealing young water contributions. These studies tend to be overlooked. Contrary to the suggestion in this comment, we encouraged sampling tritium across the full range of flow stages in catchments to avoid this potential issue.

Authors: Our position here is that we think long-term tritium sampling should be encouraged (as we recommended in the Comment) to take advantage of the changing situation with regard to remnant bomb tritium and radioactive decay of tritium, but in addition sampling should encompass the full range of streamflow rates of interest, and sampling of tritium should match that of stable isotopes as much as possible for comparison of tritium and stable isotope estimates of travel times via the SAS method. We agree that sampling tritium over high-flow events is informative and note that several papers have done that (Morgenstern et al., 2010; Cartwright & Morgenstern, 2018; Hofmann et al., 2018).

Changes: L245-248 Words changed to "For this reason, it seems unreasonable to us to consider only very detailed sampling of tritium on the scale of stable isotopes over short periods. Instead our recommendation is that more widely spaced measurements in time are included, as well as covering the full range of flow conditions so that results are not biased to low flows."

CC1: We perceive some circular reasoning in the submitted comment, which is problematic. In the comment, the TTD is assumed, to deduce what the tracer signal in the stream should be, to deduce that the tracer should not allow for discriminating young and old water in cases where the TTD is exactly as assumed (this point is not emphasized enough in the comment).

Authors: The point of the mixing calculation using the assumed TTD in Fig. 1 of the Comment is not to represent any particular system, but to show approximately the effects of lingering bomb tritium and mixing of different-age waters on tritium concentrations in stream water. The criticism appears to be that our approach is not based on real data from a real system, but from an admittedly flawed simple model. However, any real system data would only represent one example from a large number of possibilities so it would be no more valid than any other example.

Changes: No specific changes were made to our Comment because of this argument.

CC1: The real question is: is the assumed TTD realistic and accurate (especially bearing in mind the steady-state assumption)? A completely different TTD model (for example, multimodal, with both young and old water) could yield a very different perspective. More importantly, this reasoning is based on a steady-state assumption, while many more situations are possible in unsteady conditions. Virtually anything is possible in unsteady conditions, while the steady-state assumption is extremely constraining for deriving general conclusions on the link between TTDs and tracers.

Authors: There is nothing in the literature to show that a multimodal TTD model would allow better identification of old water by a seasonally-varying isotope (either stable isotopes or tritium affected by remnant bomb tritium) than a single modal model. The degree of mixing (e.g. the factor f for the EPM model) is the important factor for smoothing of seasonal variations.

Changes: Contrary to our previous response, we agree that a multimodal TTD model could allow better identification of old water (e.g. Seeger and Weiler, 2014). However, identification of the parameters of multimodal models are more difficult. No specific changes were made to our Comment because of this argument.

CC1: The presented comparison between 2H and 3H in the comment is based on different sampling strategies, which by design target different portions of the TTD. It is not so surprising that the MTTs differ, especially with a steady-state approach. Travel times are highly dynamic, even if inferred from tritium only, and the same discharge can be associated with vastly different median travel times (for instance, see figure 9 in Rodriguez et al., 2018). We thus argued that a fair comparison between the tracers needs to use tracer data sets that are as close to each other as possible, in a consistent time-varying mathematical framework.

Authors: We agree that a fair comparison between the tracers needs to use tracer data sets that are as similar to each other as possible in a consistent time-varying mathematical framework. We believe that if this is carried out on a variety of catchments it will show that tritium can identify old water components much more effectively than stable isotopes provided radioactive decay can be used for dating (as in the SH).

Changes: L198-202 The effects of sampling strategies are discussed, and some rebuttal is given here.

CC1: Other comments:

1.        L45-47: This "range" is a strong a priori assumption based on several limitations, and it is precisely what we questioned in our work.

Authors: We will change the wording here to a less prescriptive "young water".

Changes: L56-58 The wording has been changed to "Neither isotope gives any indication of a substantial presence of older water."

CC1: 2.  L48-54: We already discussed about storage S in the Weierbach catchment (section 4.4.1 in our paper). Estimating MTT from S/Q with Q the catchment runoff is wrong. The total flux through the catchment needs to be used instead. Moreover, this method too is valid only in steady-state conditions.

Authors: We agree, but this is meant as a ball-park figure not an accurate estimate.

Changes: We left this section in the Comment, but only as an indication.

CC1: To conclude, recent work suggests that there is no absolute truncation issue. The perceived "truncation" may rather have resulted from what was a too restrictive conceptualization of tritium-based TTD estimations, as also suggested by recent progress in travel time research.

Authors: There may not be a truncation issue at Weierbach Catchment, but we feel that the question is still open for other catchments. We think this is especially true for catchments in New Zealand and Australia that have high storage capacity and the very low $^3H$ activities imply MTTs of a few decades regardless of how they are calculated.

Changes: This is a conclusion. No change needed.

We fully agree with the authors of this comment on the importance of using tritium for travel time analyses, but we disagree on the fact that keeping this long-standing perception that tritium can, by default, show us this "invisible" old water will be helpful. This old water is invisible only if we choose to make it mathematically or numerically inexistent (of course there may be additional challenges of equifinality when working with a single tracer). We think that it is time to challenge our long-standing assumptions, to give up our limiting strong a priori assumptions, and to embrace the possibilities offered by the new theoretical frameworks and by the new sampling and measurement techniques. We would like to invite Stewart et al. to apply our proposed mathematical framework allowing for time variance and multimodality in TTDs to their available datasets for testing and quantifying their proposed claims. Our code is accessible online:

https://git.list.lu/catchment-eco-hydro/composite_sas_model_2h_3h_weierbach

Authors: We agree that further comparisons of stable isotope and tritium measurements on catchments (analysed by the same methods) can only be helpful to reach a fuller understanding of travel times in catchments.

Changes: No specific changes were made to our Comment because of this argument.

Cartwright, I., & Morgenstern, U. (2018). Using tritium and other geochemical tracers to address the "old water paradox" in headwater catchments. Journal of Hydrology, 563, 13-21. doi:10.1016/j.jhydrol.2018.05.060

Hofmann, H., Cartwright, I., & Morgenstern, U. (2018). Estimating retention potential of headwater catchment using Tritium time series. Journal of Hydrology, 561, 557-572. doi:10.1016/j.jhydrol.2018.04.030

Morgenstern, U., Stewart M. K., Stenger, R. 2010: Dating of streamwater using tritium in a post nuclear bomb pulse world: continuous variation of mean transit time with streamflow. *Hydrology and Earth System Sciences* 14, 2289-2301.

Rodriguez, N. B., McGuire, K. J., & Klaus, J. (2018). Time-varying storage–Water age relationships in a catchment with a Mediterranean climate. Water Resources Research, 54, 3988– 4008. https://doi.org/10.1029/2017WR021964

Seeger, S., Weiler, M.: Reevaluation of transit time distributions, mean transit times and their relation to catchment topography. Hydrology and Earth System Sciences 18, 4751-4771, 2014, doi: 10.5194/hess-18-4751-2014, 2014.

Reply to Reviewer #2 (R2)

R2: General Comments

The authors of the comment (referred to below as Stewart, 2021) provide a detailed discussion of the value of tritium analyses to constrain the travel time of streamflow, in response to a previously published paper in HESS (referred to below as Rodriguez, 2021). The comment (Stewart, 2021) specifically discusses the apparent "truncation" of travel time distributions (TTDs) derived from stable isotopes when compared to TTDs derived from tritium. This "truncation" is discussed by Rodriguez, 2021, as one of the incentives for the original study, to include both stable isotopes (i.e. $^2H$) and tritium, within the age-ranked storage selection framework (SAS). By doing that, the original paper (Rodriguez, 2021) is a very valuable contribution to the scientific literature.

Authors: We agree.

Changes: No changes needed.

R2: Specific Comments

It appears that the comment (Stewart, 2021) hinges on the interpretation of one sentence in the abstract of Rodruiguez, 2021, quoted in Stewart, 2021 on line 23:

"We conclude that stable isotopes do not seem to systematically underestimate travel times or storage compared to tritium." (Rodriguez, 2021)

More specifically, in the conclusion section of Rodriguez, 2021, the authors "conclude that the perception that stable isotopes systematically truncate the tails of TTDs may not be valid." (Rodriguez, 2021)

They continue and recommend to "compare streamflow TTD and storage from the two tracers in larger catchments where older water is expected in order to give tritium more time to decay and to better leverage its ability to point out the presence of very old water." (Rodriguez, 2021)

Considering the sentences in the conclusion section, I interpret the abstract line to be a site-specific conclusion, rather than a broad conclusion that the truncation hypothesis is "generally invalidated" (as stated by Stewart, 2021, on line 34).

The comment (Stewart, 2021) expands the interpretation of the limited conclusion by Rodriguez and states that it "does not mean that such old water does not exist in other catchments and therefore that the truncation hypothesis should be rejected for all catchments." I do not think Rodriguez intended to convey such a broad conclusion.

Recommendation

In my view, the commentary does not specifically respond to the conclusions of the Rodriguez paper, but rather to a broader interpretation that the original authors may not have intended to convey. As such, I have recommended to reject the publication of this comment as a response to the Rodriguez paper.

Authors: We feel that the issue raised in our Comment (possible underestimation of the significance of old water contributions to streamflow based on stable isotope measurements only) is worthy of discussion here, in light of the statements by Rodriguez et al (2021) that "We conclude that stable isotopes do not seem to systematically underestimate travel times or storage compared to tritium" from the abstract and "we conclude that the perception that stable isotopes systematically truncate the tails of TTDs may not be valid" from the Conclusion. As pointed out by Reviewer#1 (R1), Rodriguez et al (2021) at least give the impression that they reject the truncation hypothesis for all catchments. In addition, our Comment provides information on where and where not to expect the truncation issue

Changes: We have changed our wording in L10-14 to address our previous overstatement of the case, as pointed out in our response to the first reviewer (R1).

R2: In case the comment proceeds to publication, I have provided additional comments and suggestions below.

Specific Comments (continued)

L70 (Figure 1) It would be insightful to include the model curves for samples collected in 2010 or 2000 or 1990. For those decades, tritium may have been even less conclusive as an age tracer in streams in the Northern hemisphere. At the same time, high-frequency stable isotope studies became accessible, and was applied in northern hemisphere high-precipitation, low-ET catchments. The collective understanding of watershed response times may have been influenced by the availability of data during these decades.

Authors: We have added curves for 2000 to the graphs showing the situation with regard to tritium then.

Changes: We have added curves for the year 2000 to the graphs showing the situation with regard to tritium then, and added words (from the reviewer's comment) to the text to describe their significance (L141-145).

R2: L77: Why f=0.7? This seems arbitrary.

Authors: f = 0.7 has been found to be an effective value in studies using tritium (e.g. Morgenstern and Daughney, 2012). Calculations were also made using f = 0.5 and results were found to be similar to those obtained using f = 0.7 (these were not shown in the Comment).

The cumulative stream TTDs from Fig. 7c of Rodriguez et al. (2021) were compared with cumulative EPM curves for various values of f. The EPM curve with f=0.7 was found to give an approximate representation of the Weierbach curve. In addition, the flow-weighted cumulative stream age distributions from Fig. 7b of Visser et al (2019) are approximately consistent with the EPM(f=1) curve (otherwise known as the exponential model) as noted in their paper. Hence, f=0.7 appears to be a reasonable choice.

The value of f is very important in describing the mixing between young and old water that smooths out seasonal variations within a few years. In theory, maximum smoothing is given by f = 1 and minimum smoothing (i.e. none) by f = 0; the latter case would be very rare in streams.

Similar results would be obtained by other flow geometries that are commonly used (e.g. the gamma or the dispersion models, Stewart et al., 2017) as well as from numerical advection-dispersion models (Cartwright et al., 2018).

Changes: Fig. 1d comparing the flow-weighted cumulative stream distribution from Weierbach Catchment (Fig. 7c, Rodriguez et al., 2021) with cumulative EPM curves has been added to the Comment. This supports the choice of f=0.7. Text describing the Fig. 1d is given in L104-105.

R2: L94: "significant seasonal variation"
This is very relevant if precipitation and evapotranspiration are out of phase. Obviously, evapotranspiration is expected to remove more water in summer (when tritium concentrations in precipitation are higher) than in winter (when tritium concentrations in precipitation are lower). The degree of seasonality in evapotranspiration and tritium in precipitation, as well as the amount of mixing in the root zone, contribute to a possible bias of lower tritium concentrations in the stream, which would be interpreted as older ages.

An example (related to the cold-season-bias) is given by:

Jasechko, S.; Wassenaar, L. I.; Mayer, B., Isotopic evidence for widespread cold-season-biased groundwater recharge and young streamflow across central Canada. Hydrological Processes 2017, 31, (12), 2196-2209.

Authors: This is an important point that has been addressed in papers using tritium, by accounting for evapotranspiration losses (e.g. Stewart et al., 2007; Morgenstern et al., 2010). In Australia and New Zealand, tritium concentrations in precipitation tend to be higher in winter and early spring (Tadros et al., 2014).

The SAS method also accounts for this.

Changes: This needs to be borne in mind in individual studies, but is beyond the scope of the Comment.

R2: L160: In addition, even if older TTs derived from tritium were selectively collected during base flow conditions, that would still be evidence that the stable isotope data collected year-round fail to capture the old component in baseflow.

Authors: Agreed. Most of the Australian studies captured MTTs at a range of flow conditions and the Cartwright & Morgenstern (2018) and Hofmann et al. (2018) ones specifically estimated MTTs during flow peaks. The flow peaks were found to contain water that was a few years old.

Changes: This is in line with what the Comment is already saying. No changes needed.

R2: L163: This first point also reflects (in my opinion) a sampling bias with respect to stable isotopes and tritium to derive residence times, specifically related to the choice of the isotope applied, sampled, analyzed, interpreted, and published. Isotopic tracer studies often build on prior hydrological investigations and limited research funds are directed towards the isotopic analyses that are expected to be most valuable. Stable isotopes have been applied more often in smaller catchments with faster response times and shorter (expected) residence times, whereas tritium has more often been applied in larger river basins with longer residence times. Recent studies combining both tracers have shown that a residence time interpretation of stable isotopes may be hiding the longer tail of the distribution that can be observed by tritium.

Authors: Agreed, although there are exceptions. For example, tritium was used to show that some very small catchments in Australia had long MTTs up to 40 years (Hofmann et al., 2018; Cartwright & Morgenstern, 2016).

Changes: This is probably the prevailing opinion which Rodriguez et al. are challenging. We maintain that there are many catchments (though obviously not all) in which tritium will show older water than stable isotopes. No changes needed.

R2: However, Rodriguez shows that this is not the case in the Weierbach catchment.

Authors: Agreed

Changes: We already discuss reasons why tritium does not show older water than stable isotopes in the Weierbach Catchment. No changes needed.

R2: L183: "no issue... of interest." is not clear to me.

Authors: We will revise this statement which is poorly phrased. It is meant to say that if estimates of TTDs could be made from each one of a series of tritium measurements they would allow the time variability of the stream TTD to be determined. Whereas stable isotopes require groups of measurements to determine TTDs, tritium in principle only requires one provided radioactive decay can be used for dating and the form of the TTD can be estimated. In application, of course, the seasonal variation also needs to be considered.

Changes: L228-230 We have revised this statement

R2: L186: The short term variability of tritium is still poorly understood and so far have mostly been a nuisance for applying tritium as a short-term age tracer, although recent advances using the origin of precipitation are promising:

van Rooyen, J. D.; Palcsu, L.; Visser, A.; Vennemann, T. W.; Miller, J. A., Spatial and temporal variability of tritium in precipitation within South Africa and it's bearing on hydrological studies. Journal of Environmental Radioactivity 2021, 226, 106354.

Visser, A.; Thaw, M.; Esser, B., Analysis of air mass trajectories to explain observed variability of tritium in precipitation at the Southern Sierra Critical Zone Observatory, California, USA. Journal of Environmental Radioactivity 2018, 181, 42-51.

Authors: Agreed, this would facilitate the use of tritium as a short-term age tracer, but would be less useful for use of tritium as a longer-age tracer for which it is potentially most useful.

Changes: L231-247 This paragraph already considers the use of tritium as a short-term age tracer. No changes needed.

R2: L197: "Instead .. favoured." The design of such long term studies would also benefit from deliberately collecting tritium samples during high and low flow conditions throughout the study period to capture the time-variance of TTDs in response to hydrological conditions.

Authors: Agreed, the stream TTD response over all flow conditions is needed.

Changes: L241-245 The wording has been changed to cover the full flow response of the stream.

R2: L198: "Eventually, ... single tritium measurements." In my opinion, no single tritium measurement will be able to capture the TTD of streamflow. This statement should be removed or reworded. One tritium measurement may be able to constrain the mean travel time parameter of a TTD which shape needs to be assumed a priori.

Authors: We agree that no single tritium measurement could capture the TTD of streamflow. The last sentence is what we meant. Multiple tritium measurements are needed to capture the TTD of streamflow in different conditions.

Changes: L245-247 The wording has been changed to "Eventually, stream MTTs at particular flows will be able to be estimated from single tritium measurements provided the form of the TTD can be determined." This what the reviewer's last sentence says.

R2: Technical Comments

L19: Please also include the Van der Velde paper presenting the SAS/STOP function concept:

Van der Velde, Y.; Torfs, P. J. J. F.; Van der Zee, S. E. A. T. M.; Uijlenhoet, R., Quantifying catchment-scale mixing and its effect on time-varying travel time distributions. Water Resources Research 2012, 48, (6), W06536.

Authors: Ok

Changes: We have added this reference (L21).

R2: L45: As the travel times are a consequence of the physical and climatic characteristics of the watershed, b) should/could precede a).

Authors: We will think about this

Changes: We changed the order as suggested (L47-66).

R2: L50: I find it more useful to express catchment fluxes per unit area (in terms of m or mm, rather than volumetrically)

Authors: Ok

Changes: We did not change the streamflow units ($m^3$/year), as this is more of a personal preference (L50).

R2: L100: "using": That study didn't really "use" the variation of tritium in precipitation, but rather carefully incorporated it into the SAS modeling to avoid an old-tritium-age bias due to the strong seasonality of both ET and precipitation variation.

Authors: We will rephrase our statement.

Changes: L127-128 We have changed the wording to "Another study that determined the TTD using tritium measurements via the method of StorAge Selection functions was that of Visser et al. (2019)".

R2: L140: A thorough analysis of the (in)ability of seasonal tracer cycles to quantify mean transit times is provided by Kirchner (2016, HESS).

Kirchner, J. W., Aggregation in environmental systems–Part 1: Seasonal tracer cycles quantify young water fractions, but not mean transit times, in spatially heterogeneous catchments. Hydrology and Earth System Sciences 2016, 20, (1), 279-297.

Authors: Yes

Changes: We have put in a paragraph on the effects of aggregation (L215-219).

R2: L170: It would be helpful to provide a rebuttal to each argument listed here. (As is done for 1.)

Authors: We will consider this suggestion

Changes: We have added rebuttal arguments to items 2. and 3. (L194-208).

L186: "In addition, to" Remove comma.

Changes: L232 Done

**Citation**: https://doi.org/10.5194/hess-2021-146-RC2

IAEA/WMO (current Year). Global Network of Isotopes in Precipitation. The GNIP Database. Accessible at: https://nucleus.iaea.org/wiser

Cartwright, I., Irvine, D., Burton, C., & Morgenstern, U. (2018). Assessing the controls and uncertainties on mean transit times in contrasting headwater catchments. Journal of Hydrology, 557, 16-29. doi:10.1016/j.jhydrol.2017.12.007)

Cartwright, I., & Morgenstern, U. (2016). Contrasting transit times of water from peatlands and eucalypt forests in the Australian Alps determined by tritium: Implications for vulnerability and the source of water in upland catchments. Hydrology and Earth System Sciences, 20, 4757-4773. doi:10.5194/hess-20-4757-2016

Cartwright, I., & Morgenstern, U. (2018). Using tritium and other geochemical tracers to address the "old water paradox" in headwater catchments. Journal of Hydrology, 563, 13-21. doi:10.1016/j.jhydrol.2018.05.060

Hofmann, H., Cartwright, I., & Morgenstern, U. (2018). Estimating retention potential of headwater catchment using Tritium time series. Journal of Hydrology, 561, 557-572. doi:10.1016/j.jhydrol.2018.04.030

Morgenstern U, Daughney CJ. 2012. Groundwater age for identification of baseline groundwater quality and impacts of land-use intensification – The National Groundwater Monitoring Programme of New Zealand. Journal of Hydrology. 456–457:79–93. doi:10.1016/j.jhydrol.2012.06.010

Morgenstern, U., Stewart M. K., Stenger, R. 2010: Dating of streamwater using tritium in a post nuclear bomb pulse world: continuous variation of mean transit time with streamflow. *Hydrology and Earth System Sciences 14*, 2289-2301.

Stewart, M.K., Mehlhorn, J., Elliott, S. 2007: Hydrometric and natural tracer (18O, silica, 3H and SF6) evidence for a dominant groundwater contribution to Pukemanga Stream, New Zealand. Hydrological Processes 21(24), 3340-3356. DOI:10.1002/hyp.6557.

Stewart, M. K., Morgenstern, U., Gusyev, M. A., and Maloszewski, P.: Aggregation effects on tritium-based mean transit times and young water fractions in spatially heterogeneous catchments and groundwater systems, Hydrol. Earth Syst. Sci., 21, 4615–4627, https://doi.org/10.5194/hess-21-4615-2017, 2017.

Tadros, C. V., Hughes, C. E., Crawford, J., Hollins, S. E., & Chisari, R. (2014). Tritium in Australian precipitation: A 50 year record. Journal of Hydrology, 513, 262-273.